# Quantifying the unknown impact of segmentation uncertainty on image-based simulations

Michael C. Krygier [1], Tyler LaBonte [2,4], Carianne Martinez[2], Chance Norris [3], Krish Sharma[2], Lincoln N. Collins [1], Partha P. Mukherjee[3] & Scott A. Roberts [1✉]

Image-based simulation, the use of 3D images to calculate physical quantities, relies on image segmentation for geometry creation. However, this process introduces image segmentation uncertainty because different segmentation tools (both manual and machine-learning-based) will each produce a unique and valid segmentation. First, we demonstrate that these variations propagate into the physics simulations, compromising the resulting physics quantities. Second, we propose a general framework for rapidly quantifying segmentation uncertainty. Through the creation and sampling of segmentation uncertainty probability maps, we systematically and objectively create uncertainty distributions of the physics quantities. We show that physics quantity uncertainty distributions can follow a Normal distribution, but, in more complicated physics simulations, the resulting uncertainty distribution can be surprisingly nontrivial. We establish that bounding segmentation uncertainty can fail in these nontrivial situations. While our work does not eliminate segmentation uncertainty, it improves simulation credibility by making visible the previously unrecognized segmentation uncertainty plaguing image-based simulation.

[1] Engineering Sciences Center, Sandia National Laboratories, Albuquerque, NM, USA. [2] Applied Machine Intelligence and Application Engineering, Sandia National Laboratories, Albuquerque, NM, USA. [3] School of Mechanical Engineering, Purdue University, West Lafayette, IN, USA. [4] Present address: Machine Learning Center, Georgia Institute of Technology, Atlanta, GA, USA. ✉email: sarober@sandia.gov

Image-based simulation is the process of performing quantitative numerical calculations, such as 3D finite-element simulations, on geometries constructed directly from 3D imaging techniques, including X-ray computed tomography (CT) and scanning electron microscopy. Image-based simulation has become a crucial part of modern engineering analysis workflows, alongside integrated computational materials engineering (ICME)[1] and digital twin[2–5] efforts. It has also been adopted by many disciplines, including medical imaging for patient treatment[6–10], improving the manufacturing process of batteries[11–14], composite material development[15–18], biological physics[19], neuroscience[20,21], and geomechanics[22].

Image-based simulation traditionally involves three steps. The first step, image segmentation, is the classification of each voxel (3D pixel) in the image to a distinct class or material (Fig. 1). For instance, the blue and purple regions in Fig. 1b represent two different materials captured in the image in Fig. 1a. The second step is the reconstruction of a computational domain from the image segmentation. The third step is the numerical simulation of physics quantities on this reconstructed domain. The outcome of this image-based simulation (Fig. 2c) is a single value for the physics quantity of interest (Fig. 2d).

For this process (Fig. 2b), traditional image segmentation approaches involve manual segmentation, whereby a person applies a combination of image filtering techniques (e.g., smoothing, noise removal, contrast enhancement, or non-local means filters) and segmentation algorithms (e.g., simple thresholding, watershed, or multi-Otsu thresholding algorithms) to segment the image. However, manual segmentation is fraught with irreproducibility and person-to-person variability. While the tools themselves are scientifically sound, the way that people deploy them makes it more of an art than a science. Two qualified individuals performing segmentation on the same image are likely to choose a different combination of filtering and segmentation techniques (or parameters for those techniques) leading to different segmentations[23–26]. While to the human eye, one algorithm may be subjectively better than another algorithm for a given image, each segmentation is a plausible and valid segmentation. This suggests that even when a clear segmentation is achieved (e.g. the black curve in Fig. 1b), it can never be verified as the most correct segmentation. The range of possible segmentations, accounting for all tools and variables, represents the range of image segmentation uncertainty (the yellow regions in Fig. 1b), within which the correct answer will be found. However, because there are infinite combinations of manual segmentation algorithms and parameters, it is difficult to fully characterize segmentation uncertainty using manual segmentation approaches.

Machine learning techniques, and convolutional neural networks (CNNs) in particular, have revolutionized image segmentation by alleviating three main disadvantages in manual segmentations[27]. First, manual segmentation is a labor-intensive task, and CNNs help to remove this burden. Second, CNNs can often achieve better accuracy than humans[28,29], even when trained on imperfect manual segmentations. Third, CNNs produce consistent segmentations that are deterministic at inference, generating reproducible results over many images of a similar domain. Because of these advantages, CNNs have gained immense popularity for image segmentation in a variety of applications, including in energy storage[30], materials analyses[31–33], and medical diagnosis[34,35].

However, although CNN-based segmentation has many advantages, it is not without segmentation uncertainty. Strikingly, the same problem that plagues manual segmentation also plagues CNN-based segmentation. For instance, each CNN is designed using different stencils, varying number of layers, and parameterizations, similar to the application of manual segmentation algorithms. In addition, image artifacts, noisy input images, and imperfect manual segmentations used for model training introduce variability in inference samples, resulting in segmentation uncertainty. Therefore, it is rational to ask—how reliable are CNN-based segmentations? In the medical field, such reliability concerns are preventing neural networks from being fully utilized in a clinical setting[36,37] and have led to a grand challenge for the quantification of uncertainties in biomedical image quantification[38]. At present, the most common way to address CNN-based segmentation uncertainty is through Monte Carlo dropout networks (MCDNs)[39], which have been applied in numerous disciplines[36,40–44]. However, MCDNs do not inherently capture segmentation uncertainty within the CNN design; instead, probing segmentation uncertainty through a stochastic sampling of dropout layers is an established approach, albeit with questionable statistical validity[45]. However, LaBonte et al.[46] has developed a Bayesian CNN (BCNN) that measures uncertainty in the weight space, resulting in statistically justified sample inferences for segmentation uncertainty quantification. While these advances indicate that there is uncertainty in image segmentation and provide a method to visualize it, they provide no quantitative method to propagate the image segmentation uncertainty through physics simulations.

Because image-based simulations use segmented images, uncertainty in those image segmentation will necessarily lead to uncertainty distributions in the physics quantities predicted by image-based simulations. For high-consequence applications, quantifying uncertainty distributions derived from image segmentation uncertainty leads to a credible image-based simulation workflow. To date, this concept has received very little attention in the literature. Very recently the impact of segmentation uncertainty on physics quantities has been acknowledged for manual segmentation[9,26,47]. However, because these works focus exclusively on manual segmentation, the scope of their proposed solutions is limited to relatively subjective method-to-method comparisons, which are both irrelevant for CNN-based segmentation and potentially overlook the more comprehensive question of how segmentation uncertainty directly impacts the physics simulations. There is clearly a need for a more consistent and systematic approach to quantify the uncertainty distributions of physics quantities resulting from segmentation uncertainty and preferably an approach that makes use of more modern and objective CNN-based image segmentation techniques.

Herein, we address this challenge by presenting a systematic method of quantifying segmentation uncertainty and propagating that uncertainty through image-based simulations to create uncertainty distributions on predicted physics quantities, as illustrated in Fig. 2e–j. Our efficient quantification of uncertainty in

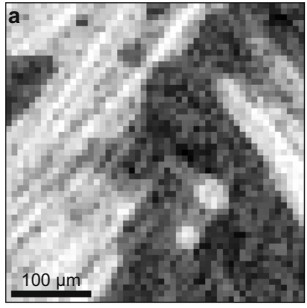 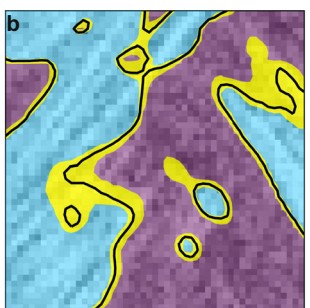

**Fig. 1 Illustration of image segmentation and segmentation uncertainty.** A grayscale image (**a**) is segmented into white (blue region) and black (purple region) classes in (**b**), with black curves denoting one possible interface boundary between classes. The yellow region is a visual representation of the segmentation uncertainty that results from all possible image segmentations.

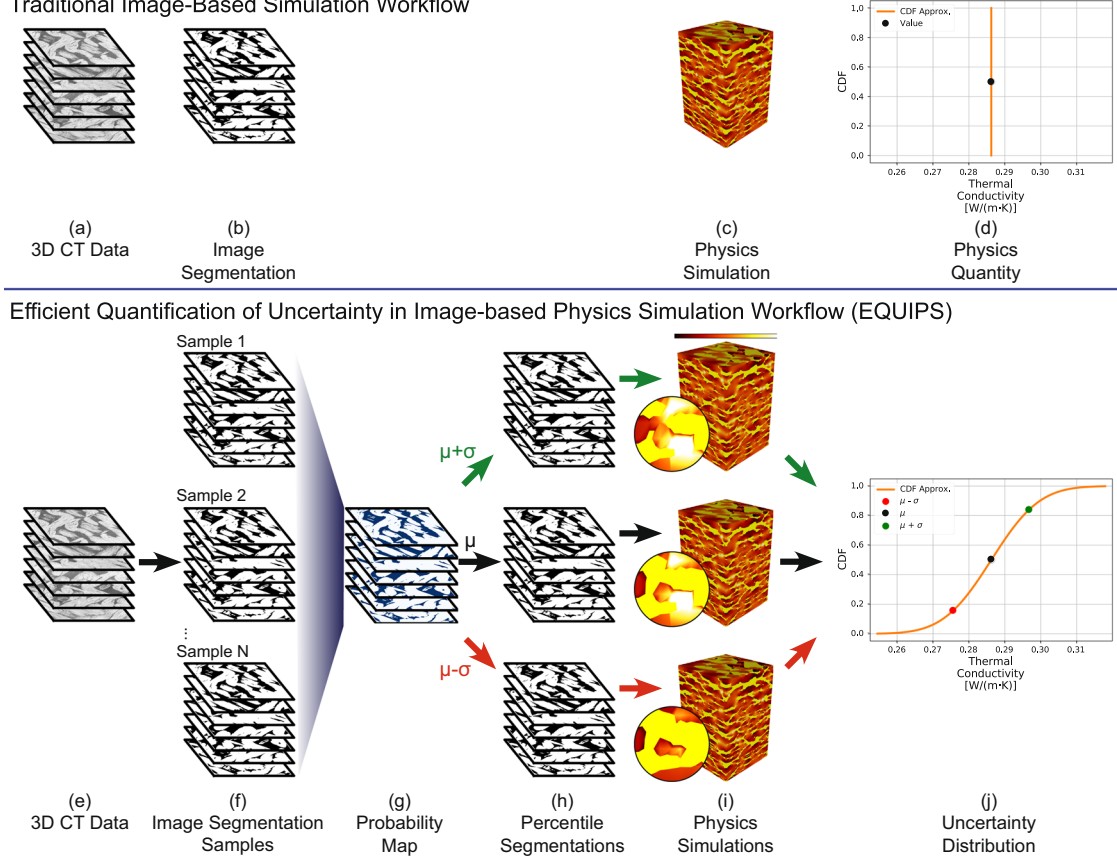

**Fig. 2 Traditional image-based simulation workflow (top) versus proposed efficient quantification of uncertainty in image-based physics simulation (EQUIPS) workflow (bottom).** A traditional image-based simulation workflow (top) converts 3D images (**a**) into image segmentations (**b**) using manual or CNN-based algorithms, then performs a numerical simulation on the reconstructed segmented image domain (**c**) to calculate a deterministic physics quantity (**d**). In EQUIPS (bottom), segmentation uncertainty is calculated from 3D images (**e**) by creating many image segmentation samples (**f**) and combining them into a probability map (**g**). The probability map is thresholded at different percentile values to construct percentile segmentation domains (**h**), each of which are used to perform physics simulations (**i**) whose output is combined into an uncertainty distribution (**j**), represented as a cumulative distribution function (CDF) for that physics quantity. The colors in (**i**) show localized heat flux in a thermal conductivity simulation on a relative (arbitrary) color scale.

image-based physics simulation (EQUIPS) workflow is general in that it is agnostic to the physical system (i.e., material) being studied, the image-segmentation approach, the method for performing the image-based simulation, and the physics quantities. In our workflow, EQUIPS employs both MCDNs and BCNNs to perform image segmentation to quantify segmentation uncertainty, with both approaches creating a set of image segmentation samples (Fig. 2f). These samples are combined to create a single probability map (Fig. 2g) that objectively represents the probability that a certain voxel is in the segmented material class. To explore the impact of segmentation uncertainty on physics quantities, we threshold the probability map at certain percentiles to obtain percentile segmentations (Fig. 2h), which are then used to perform multiple physics simulations (Fig. 2i) and calculate uncertainty distributions in physics quantities of interest (Fig. 2j). In the next section, we describe this approach in more detail, demonstrating the value of EQUIPS by quantifying the effect of segmentation uncertainty on physical quantities in three distinct exemplars: woven composites, battery electrodes, and a human torso.

## Results
We begin by illustrating the EQUIPS workflow for quantifying segmentation uncertainty and propagating it to physics

simulations on the exemplar of a woven composite material (Fig. 2). We train a BCNN to segment 3D grayscale CT images (Fig. 2e). The output of the network is a softmax layer that, when thresholded at 0.5, creates a binary representation of voxels that are inside the segmented class. Next, we Monte Carlo sample the network to generate $N$ unique image segmentation samples (Fig. 2f). Each image segmentation sample represents a valid inference through the model, with each sample probing the image and model uncertainty stochastically.

The probability map ($\epsilon$, Fig. 2g) is the per-voxel mean of the $N$ image segmentation samples (Fig. 2f), as mathematically defined in the "Methods" section. Intuitively, the probability map represents the probability that a voxel is in a segmented class. As a result, this per-voxel probability distribution can be probed by thresholding the data with the desired probability value, creating a binarization that represents that probability threshold. For example, thresholding at $\epsilon \geq 0.20$ generates a binarization containing all voxels that have at least a 20% probability of being in the segmented class. Conceptually, this process can be thought of as probing the cumulative distribution function (CDF) of the segmentation uncertainty, where the chosen probability value represents a percentile in that distribution. The result of this process produces a percentile segmentation, illustrated by each of the image stacks in Fig. 2h. For a multi-class problem, this workflow is repeated on each class individually.

EQUIPS then probes the segmentation uncertainty at three probability values: $\mu-\sigma$, $\mu$, and $\mu+\sigma$, which we will call the standard segmentations (Fig. 2h). Here, $\mu$ is the mean, 50.0 percentile, and $\sigma$ is the standard deviation away from the mean, 15.9 and 84.1 percentiles at $\mu-\sigma$ and $\mu+\sigma$, respectively. We chose these particular percentile values to quickly model the uncertainty distribution in a physics quantity using a characteristic distribution. However, any percentile from the probability map could be used to generate an image segmentation representing that probability value.

For each standard segmentation, we perform a physics simulation to predict the physics quantities (Fig. 2i). Each inset in Fig. 2i highlights a region of the domain geometry that undergoes significant alterations. These geometry changes impact the physics quantity behavior and, as a result, these changes manifest in the physics quantity uncertainty distribution. For example, in the $\mu-\sigma$ case, the middle inset shows a material region that is disconnected from the surrounding region and therefore has low heat flux. In contrast, the $\mu+\sigma$ case shows a material region that is fully connected to its neighbors and therefore has a much higher heat flux.

Throughout this work, we focus on a Normal (Gaussian) distribution as the characteristic distribution to rapidly approximate a physics quantity uncertainty distribution using the fewest possible simulations. Only the three standard segmentation simulations are necessary to specify this characteristic distribution. The physics quantity evaluated using the 50.0 percentile segmentation provides the distribution mean, while the 15.9 and 84.1 percentile segmentations provide the standard deviation. The curve in Fig. 2j is the CDF of the characteristic distribution estimate calculated using this method while the red, black, and green points are the standard segmentation values. This characteristic CDF estimates the uncertainty distribution of a physics quantity as a result of segmentation uncertainty. If the characteristic distribution estimate fits the physics quantity data points well, then the calculation of additional percentile segmentations is likely unnecessary. However, if the characteristic distribution is poor (i.e. the unknown distribution is non-Normal), then additional percentile segmentations are required to capture the underlying physics quantity uncertainty distribution.

In the following discussion, we present three exemplar problems demonstrating the propagation of segmentation uncertainty from image segmentation through physics simulations to physics quantities. In each of these exemplars, simulation conditions and model parameters are held constant to draw attention to the impact that segmentation uncertainty has on the uncertainty distribution of physics quantities.

**Woven composite**. In this exemplar, we use 3D CT scans of a woven composite material (Fig. 2e) to simulate thermophysical quantities relevant to its application as a thermal protection system for an atmospheric entry vehicle. In these scans, we segment fabric yarn material from the resin phase. We focus on three physics quantities: fabric volume fraction, effective thermal conductivity, and fluid permeability.

We introduce an uncertainty map (Fig. 3a) to visualize and quantify uncertain regions within the image. We use the probability map $\epsilon$ to calculate the voxel-wise uncertainty map using the Shannon entropy. A voxel is most uncertain when it can be assigned to any class with equal probability. Regions with high uncertainty tend to occur at the boundaries between material phases, as these are the voxels in the grayscale image that are the most ambiguous for segmentation. While it is typical that the uncertainty is concentrated near material phase boundaries, it does not necessarily have to be so; CNNs can identify regions that are nominally within a material but whose grayscale image values suggest ambiguity in its assignment to that material. Additionally, the standard segmentation contours ($\mu-\sigma$, $\mu$, $\mu+\sigma$) are overlaid on the probability map as red, blue, and yellow lines, to illustrate their respective class interface boundaries in the probability map using a zoomed-in image section (Fig. 3b).

This segmentation uncertainty affects different physics simulations differently, as we illustrate in Fig. 3c for volume fraction and effective thermal conductivity, with both physics quantities normalized to their mean values. The volume fraction is the natural outcome of image segmentation, as it does not require a physics simulation to calculate and therefore has no potential amplifications or nonlinear interactions with the physics model. Thermal conductivity, however, is more sensitive to small changes in the geometry, as previously illustrated in the insets of Fig. 2i. The addition of only a few voxels to a high conductivity material may connect previously isolated regions, adding a new pathway for heat conduction and drastically increasing the effective thermal conductivity. Fig. 3c shows this to be the case, as the uncertainty distribution in thermal conductivity is nearly twice the uncertainty in volume fraction, with the physics interactions amplifying the uncertainty distribution.

In spite of their varying uncertainty magnitudes, both thermal conductivity and volume fraction follow a Normal distribution. We performed eleven simulations to produce the data points in Fig. 3c. Calculated using only the standard segmentation results (black and green markers), the CDF (solid curves) adequately captures the propagation of segmentation uncertainty to these two physics quantities. We found that the extra percentile segmentation simulation results (red markers), which were not used to calculate the CDF, fall on the CDF curves. Thus, the physics quantity uncertainty distribution that follows this Normal distribution can be adequately represented with only three physics simulations. This outcome validates our choice of this characteristic distribution.

However, three physics simulations are not adequate for fluid permeability, which is better approximated by a beta distribution (Fig. 3d). We hypothesize that the non-Normal uncertainty distributions are often the result of nonlinear physics interactions. For instance, the characteristic Normal distribution suggests that it is possible to have negative permeability, which is physically impossible. In this scenario, we needed to perform 10 percentile segmentations and physics simulations to acquire the uncertainty distribution.

**Graphite electrodes in lithium-ion batteries**. For the next exemplar, we studied two visually distinct CT scans of graphite battery electrode microstructures: Electrode I (E1, Fig. 4a) and Electrode II (E2, Fig. 4e), where the segmented class describes the particle phase. This exemplar was selected to explore the role that image quality has on segmentation uncertainty. Subjective visual inspection of these two images suggests that the E2 image is sharper, with more visually distinct particles, while E1 has much less contrast between the particle and void phases and has blurrier edges. This subjective assessment is confirmed with the blind/referenceless image spatial quality evaluator (BRISQUE)[48,49], where E1 scores a 159 and E2 scores a 125 (where a smaller BRISQUE score indicates superior perceptual image quality). Additionally, the grayscale histograms of each image (Fig. 4b, f) confirm this assessment, with E1 showing a much broader and single-mode distribution. It would be quite difficult to choose a simple threshold value for image segmentation based solely off of the histogram for E1. Given the qualitative and quantitative image quality differences between these two electrodes, we hypothesized that a credible segmentation uncertainty quantification approach

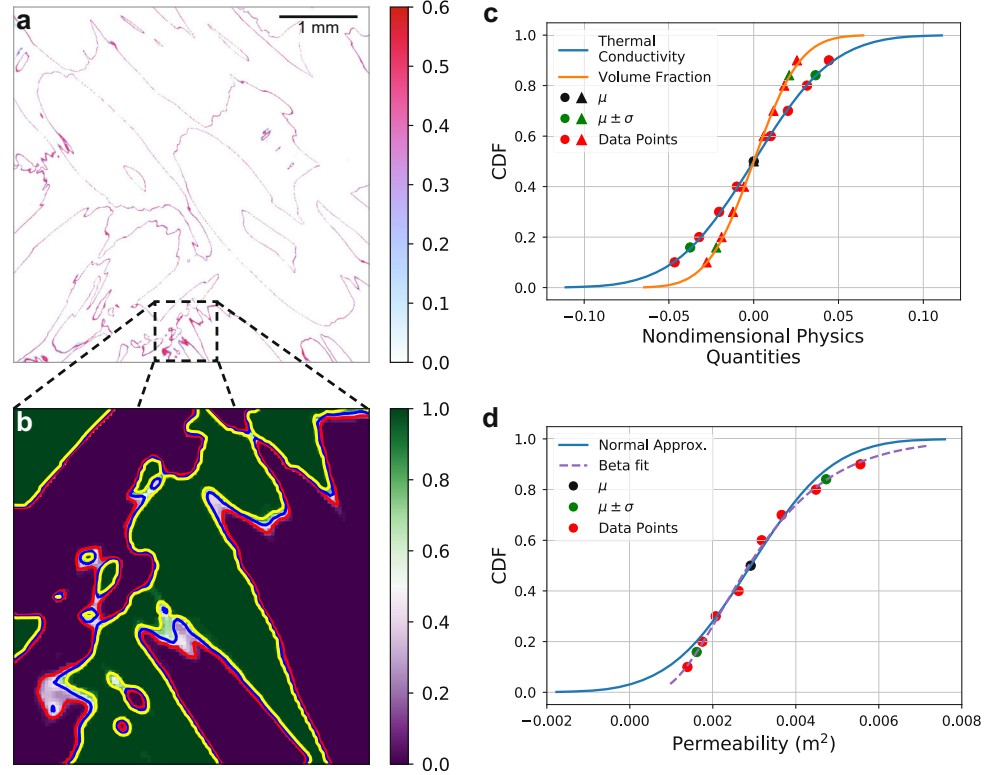

**Fig. 3 Segmentation uncertainty in a woven composite material. a** Graphical 2D representation of uncertainty, with the highest uncertainty occurring near material boundaries. **b** Visualization of the standard segmentation contours in a zoomed-in region of the probability map that uses the same slice plane shown in (**a**), where the red, blue, and yellow contours are the standard segmentations, $\mu - \sigma$, $\mu$, and $\mu + \sigma$, respectively. **c** CDF for two physics quantities, thermal conductivity (circles) and volume fraction (triangles), highlighting that some physics quantities are more sensitive to segmentation uncertainty than others. **d** CDF of fluid permeability, which exhibits a non-Normal distribution.

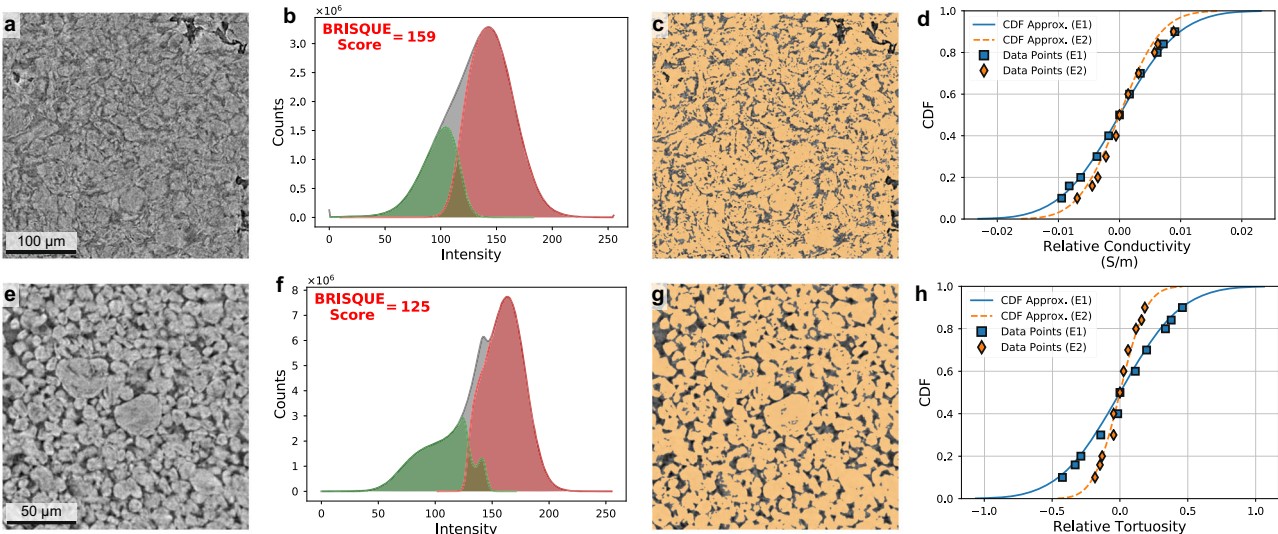

**Fig. 4 Segmentation uncertainty in two graphite electrodes for lithium-ion batteries.** Electrode I (**a**–**c**) and Electrode II (**e**–**g**). **a**, **e** 2D slices of the 3D X-ray CT image. **b**, **f** Histogram of voxel intensities for the whole image (gray), for voxels segmented as particle (red), and for voxels segmented as void (green). The BRISQUE image quality score is overlaid on the histograms and quantitatively measures the perceptual quality of an image, with a lower score indicating superior image quality. **c**, **g** Segmentation in transparent orange overlaid on the original 2D image. **d**, **h** Uncertainty distribution for both electrode images, with electrical conductivity in (**d**) and tortuosity in (**h**), both relative to their mean values.

should report a wider uncertainty distribution for a lower quality image such as E1.

It is worth noting that the CNN is performing a non-trivial segmentation. Gray histograms in Fig. 4b, f show the overall distribution of grayscale values for each image. For simple thresholding, the histogram would ideally exhibit a bimodal distribution, and the valley between the two peaks would be chosen as the threshold value. The red and green histograms represent the grayscale values that the network assigned to the particle and void classes, respectively. If a simple threshold had been used, then there would be a sharp vertical delineation between these two histograms. Instead, there is a region of grayscale values showing significant overlap between particle and void phase assignments. The overlap highlights the fact that the CNN is not simply a method for calculating a grayscale threshold; it is actually learning shapes and features in the image. Interestingly, the overlapping region is larger for the lower quality image (E1), confirming that the segmentation of this image is more difficult. Finally, the accuracy of the CNN segmentation is highlighted in Fig. 4c, g which show the $\mu$ percentile segmentation overlaid on the grayscale image. While the segmentation is accurate for both images, the accuracy is qualitatively better for the higher quality E2 image.

We chose to characterize two physics quantities relevant to battery performance: effective electrical conductivity and effective tortuosity, whose respective uncertainty distributions are shown in Fig. 4d, h relative to their mean values. The characteristic Normal distribution captures the uncertainty in electrical conductivity well for both electrodes (Fig. 4d), with only the tails deviating slightly from the Normal distribution values. However, the tortuosity uncertainty distribution follows the characteristic distribution less closely (Fig. 4h). For both E1 and E2, the characteristic distribution overestimates the data in the distribution tails.

Even more important are the relative uncertainty distributions between E1 and E2 for both physics quantities. As expected, the lower-quality image E1 has higher uncertainty than E2 for both the electrical conductivity (22.2% higher standard deviation) and the tortuosity (58.7% higher standard deviation). This discrepancy is direct evidence that EQUIPS is capturing the uncertainty associated with image quality (both noise and edge sharpness) and is propagating the uncertainty through the governing physical equations to the physics quantities. Finally, this exemplar provides credibility to EQUIPS, as our results intuitively correlate to both the visual assessment of the quality of the two CT scans (as blurry and sharp, respectively) and their verified BRISQUE scores.

**Human torso**. In our final exemplar, we focus on the segmentation and simulation of both the spine and aorta from a human torso. A MCDN is used for this exemplar rather than the BCNN because of its ability to perform multi-class segmentations, but also to highlight the flexibility of EQUIPS, which can work with a variety of segmentation algorithms. We show the spine and aorta as green and orange, respectively, in 2D CT stacked slices to illustrate their location in the 3D image (Fig. 5a). A 3D representation of the $\mu$-percentile segmentation of the spine and aorta class combination is shown in Fig. 5b. While simulations for each of these component organs are performed independently, this exemplar highlights the ability of our MCDN-based approach to perform multi-class analysis, segmenting and assessing the uncertainty of multiple organs simultaneously.

We begin with the spine, where the effective axial Young's modulus is the scalar physics quantity. Full-field vertical and lateral displacements are visualized in Fig. 5c, d, respectively.

Because of the intricately complex structure of the spine, which is comprised of irregular bone segments (vertebrae) connected by small joint regions, this exemplar shows more non-trivial nonlinear solution results than the previous exemplars. This connectivity results in a complex load path that admits numerous rotations of individual vertebrae, with the second vertebra from the bottom showing the highest vertical displacement and the fifth vertebra giving a negative displacement (net downward movement) in the anterior portion. The complex interconnections also result in a lateral bowing displacement on the same order of magnitude as the applied vertical displacement.

The complex spinal load path and displacements predictably lead to complex interactions between segmentation uncertainty and physics simulations, with the modulus distribution estimates shown in Fig. 5e. To accurately resolve the non-trivial physics uncertainty distribution, we ran 25 percentile segmentation simulations rather than the 10 used in previous exemplars. Clearly, the Normal distribution is a particularly poor fit to the simulation data. A half-Cauchy distribution fits much better because the additional voxels segmented as bone above the 75th percentile appears near the free-moving joint regions of the spine, significantly stiffening up the load path. In this case, not all bone is created equal as it contributes to the effective stiffness of the spine.

The aorta investigation focuses on the risk of abdominal aortic aneurysms, quantified by the ratio of the vessel wall area above a threshold shear stress ($\delta_\tau$, yellow regions in Fig. 5f)[50]. A second physics quantity that we explore is the ratio of outlet flow rates between the side vessel branches and the main aorta (Fig. 5g). Unlike the previous examples, the physics quantities in this exemplar are transient, as they are driven by a time-dependent pulsatile pressure gradient (gray curve in Fig. 5g).

Because of the relatively high flow rates, it takes at least four full pulse periods to reach a pseudo-steady-state flow solution, as shown by the flow-rate ratio in Fig. 5g. Instead of CDFs, which we have shown for scalar and steady-state physics quantities previously, we represent the uncertainty in this flow-rate ratio as a set of transient curves for each of the three standard segmentations, with the orange shaded region representing the $\mu \pm \sigma$ uncertainty estimate. Larger percentile segmentations lead to a higher flow-rate ratio, as the added aorta voxels more significantly increase the available vessel flow area for the smaller side-branch vessels than the vertical main branch. Inertial effects of the flow additionally lead to both smoothing and delay in the peak of the flow-rate ratio compared to the applied inlet pressure.

Of all of the physics quantities up to this point, the aorta wall-shear-stress threshold ratio, $\delta_\tau$, shows the most complex (and initially counter-intuitive) interactions with segmentation uncertainty. Figure 5h shows the calculated ratio for each of the 11 calculated percentiles over the final pressure pulse, with line color representing the simulation's percentile segmentation and thick solid curves representing the standard segmentations. While the behavior and trends during the peak of the flow, where the ratio is the highest, are monotonic and intuitive, the ratio later in time gradually becomes increasingly non-monotonic with the percentile segmentation. Surprisingly, the $\mu$-percentile segmentation's ratio crosses outside the bounds calculated from the ($\mu \pm \sigma$)-percentile segmentations. This result is unexpected but shows the sensitivity of propagating segmentation uncertainty in image-based simulations to calculated physics quantities, particularly when the physics model is complicated and nonlinear, such as the Navier–Stokes equations at a high Reynolds number.

To help further illustrate this critical result, the region between the ($\mu + \sigma$)- and ($\mu - \sigma$)-percentile segmentation ratio curves are shaded dark gray and the region between the 10- and 90-percentile segmentation curves is shaded light gray. At the ratio

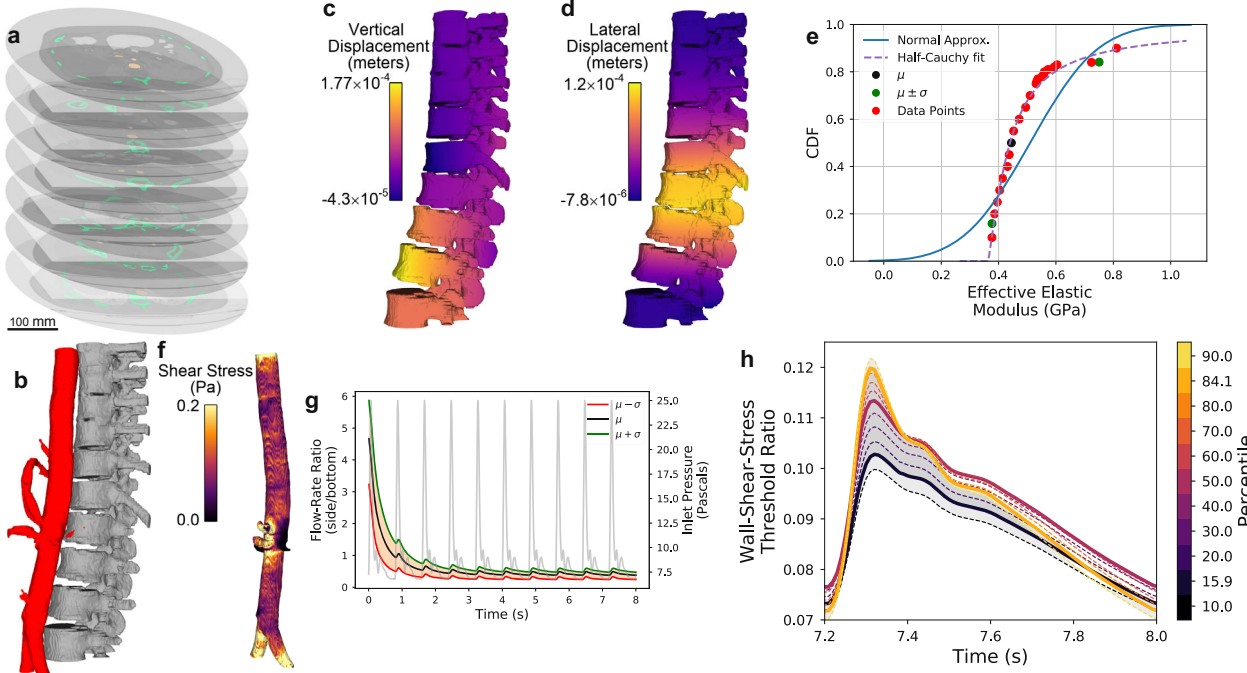

**Fig. 5 A multi-class, multi-physics uncertainty analysis of a CT scan of a human torso. a** 2D slices of the CT image labels for the spine and aorta overlaid in green and orange, respectively. **b** A 3D visualization of the $\mu$-percentile segmentation of both classes (spine and aorta). Visualization of spine axial compression simulation showing **c** vertical displacements, **d** lateral displacements, and **e** the resulting uncertainty distribution in the effective Young's modulus, including distribution estimates. **f** Visualization of the wall shear stress in the aorta resulting from a flow simulation. **g** Prescribed aorta inlet pressure profile (gray) overlaid on the time-dependent outlet side-to-bottom flow rate ratio, including uncertainty bounds. **h** Aorta wall-shear-stress threshold ratio for each percentile segmentation, with the standard segmentations shown as thick solid curves. The dark gray-shaded region emphasizes the curves bounded between ($\mu + \sigma$)- and ($\mu - \sigma$)-percentile segmentations, whereas the light gray-shaded region highlights the space between the 10- and 90-percentile segmentations.

peak, the behavior is intuitive (i.e., monotonic) with the shaded regions enveloping all their respective curves. However, later in time, the ratio gradually becomes increasingly non-monotonic with percentile segmentation. At $t = 7.8$ s the 10- and 90-percentile segmentation ratios fail to bound other percentile segmentation ratios and ultimately intersect each other moments later. Moreover, each ratio becomes progressively intertwined with others, such that it becomes difficult to predict how a new percentile segmentation ratio calculation would fall on this graph.

The $\delta_\tau$ curves portrayed in Fig. 5h emphasize the caution required in bounding segmentation uncertainty using only a few simulations. In the first two exemplars, the results are monotonic with increasing percentile segmentations, and the approximation of the uncertainty distribution using a CDF is representative. However, that does not have to be the case with more complicated models. For example, simply using ($\mu \pm \sigma$)-percentile segmentations to bound the segmentation uncertainty not only fails to encompass all of the intermediate percentile values, it also fails to capture the mean behavior. While this does not invalidate our approach to segmentation uncertainty quantification, it does urge caution in blindly performing simulations for only the standard segmentations for a new exemplar without first checking more intermediate segmentations.

## Discussion

In this work, we developed EQUIPS, a framework for quantifying image segmentation uncertainty and propagating that uncertainty to physics simulations to create uncertainty distributions for physics quantities. We demonstrated the general value and flexibility of EQUIPS in a multi-disciplinary context by using three carefully chosen exemplars. First, we used a woven composite

material to show that EQUIPS can capture varied uncertainty distributions for different physics quantities, including both Normal and non-Normal distributions. Second, we used a battery electrode to show that lower-quality images have higher segmentation uncertainty than higher-quality images. Third, we applied EQUIPS to both time-dependent and multi-class datasets in a medical context. In doing so, we further discovered that physics simulations can amplify or suppress segmentation uncertainty in both linear and non-linear manners, leading to unpredictable results and requiring caution when simply propagating lower and upper segmentation uncertainty bounds to physics quantities.

We use a characteristic distribution, chosen as Normal distribution, to quickly approximate the underlying uncertainty distribution of a physics quantity using only the standard segmentations, which minimized the number of physics simulations necessary to recover a physics quantity uncertainty distribution from the probability map. If the characteristic distribution fits the data points well, then additional image-based simulations are not necessary. However, when the characteristic distribution fails to adequately capture individual physics quantity data points, then the probability map must be probed further. Using this rapid approach, we show that estimating the uncertainty distribution using the characteristic distribution works wells for some physics quantities. For example, thermal and electrical conductivity calculated from image-based simulations of the woven-composite material and battery electrodes both fit the Normal distribution nicely. However, we also show that the characteristic CDF for permeability of the woven composite and tortuosity of the battery electrodes poorly matches the distribution tails. Thus, EQUIPS is promising but is currently system- and physics-quantity-dependent. Nevertheless,

the standard segmentations provide an excellent starting point for quantifying and propagating segmentation uncertainty to a physics quantity in image-based simulations.

Furthermore, in the medical exemplar, we show that the uncertainty distribution is non-trivial and can result in non-monotonic results. This is direct evidence that slightly changing the image segmentation can have drastic changes in the calculated physics quantities. Consequently, we caution against ad-hoc approaches[9,26,47] at bounding uncertainty using only a handful of segmentations. Even within EQUIPS, the validity of the characteristic distribution must first be verified for a new exemplar using more percentile segmentations. However, we conjecture that, once the monotonicity and shape of the resulting distribution are confirmed for a specific material class and simulation type, a simpler approach is warranted for future similar images.

In this work, we use CNNs to probe image segmentation uncertainty through Monte Carlo sampling of the network, but CNNs are not a requirement in our workflow. Alternative methods of generating multiple image segmentation samples include the probing of manual segmentation[9,51] and Bayesian Markov chain Monte Carlo[52] algorithms. While we believe that CNN-based segmentation approaches are generally superior to these other algorithms, some of these approaches may already be in heavy use and their replacement with CNNs would be time-consuming. To quantify segmentation uncertainty using alternative segmentation algorithms, each of their output segmentations could be combined into a probability map by replacing the CNN model inferences in Fig. 2f with the multiple-image segmentations produced by these algorithms. For manual image segmentation, probability maps could be generated by having multiple subject matter experts each segment the image, which would then be combined into a probability map. However, there is a potentially unreasonably large overhead in having enough experts segment images to generate a statistically relevant probability map. In contrast, segmentation approaches such as random walker[53] and trainable Weka segmentation[54] return a voxel-wise probability map that can replace the probability map in our workflow (Fig. 2g), eliminating the need to generate multiple images segmentation samples. Although we have not demonstrated these approaches, replacing the CNN in our workflow with any of these methods would be relatively straightforward, and this plug-and-play feature of EQUIPS expands our workflow's applicability beyond the requirements of neural networks.

In this exploration of segmentation uncertainty, we did not differentiate between the effects of aleatoric uncertainty (due to probabilistic events) and epistemic uncertainty (due to uncertainty in system information). Instead, we explored the combined effects in propagating segmentation uncertainty to physics predictions from image-based simulations. Furthermore, our segmentation uncertainty investigation focuses on segmentation uncertainty on a per-voxel level. However, it could be advantageous to understand correlations between the uncertainties of neighboring voxels. Future work on segmentation uncertainty can explore both of these avenues and many more.

In discussing the human torso, we introduced the concept of a multi-class image, one that includes more than one segmented phase. However, in our physics calculations for the torso, each segmented class was treated independently. A generalized approach to probing the segmentation uncertainty of multi-class images, where the physics calculation uses all of the classes in the image, is available in the supplementary information.

As we have shown, plausible changes to segmentations of image data can have a significant influence on physics quantities. This startling revelation suggests that segmentation uncertainty must be included in future image-based simulations to quickly determine whether the underlying governing equations are

sensitive to image segmentation or image noise. Image-based simulation workflows with established credibility will spark future innovation in numerous new applications, including reducing drug-development time[55], developing patient-specific cancer treatments[7,9,56,57], in digital twins, and in qualifying additively manufactured components. While image collection and processing techniques will only improve, this work will set the foundation for realizing the impact that image segmentation uncertainty has on the uncertainty distributions of physics quantities from image-based simulation.

## Methods

**Convolutional neural networks**. In this work, we use both MCDNs[39] or Bayesian convolutional neural networks (BCNNs)[58] to automate the task of CT segmentation. Each model is implemented with a V-Net architecture[27] commonly used for 3D image segmentation. The main structural components are described as follows: a V-Net first downsamples the input CT scan volume through multiple resolutions, with each resolution containing convolutional filters ultimately resulting in a larger receptive field. Copies of the resulting outputs of each of these downsampling layers are passed on via skip connections to an upsampling half of the network as features in order to minimize information loss that results from downsampling. After symmetric upsampling, using deconvolutional layers of complementary size to the paired downsampling layer, the resulting output from all of the convolutional layers is of the original input size. Finally, a voxel-wise sigmoid or softmax activation function is applied to the volume, resulting in real-valued output for each voxel with a value between 0 and 1 for each class.

Segmentation uncertainty is inherently captured in both networks. In MCDNs, dropout layers that remove the output from a randomly selected subset of nodes in the neural network during each calculation are active during training (to mitigate overfitting) and during inference (to introduce variance in the model's predictions). The standard deviation over many model inferences for the same input is a measure of the model's uncertainty. In contrast, BCNNs frame the process of learning the network parameters as a Bayesian optimization task and, instead of point estimates, they learn a set of parameters—in most cases the mean and variance that define a Normal distribution over each network weight. Intuitively, the magnitude of the standard deviation of every weight's distribution captures how uncertain the network is for that specific weight. By sampling the network multiple times on the same input, we effectively sample the network weight space. The uncertainty is then quantified in the output space of the network by calculating the variability in the voxel-wise sigmoid outputs over multiple samples of the weight space.

The BCNN[46] combines the concepts of a BNN, the V-Net architecture, and several deep learning advances in training paradigms to produce a model capable of binary segmentation. The BCNN places a Gaussian prior distribution over each of the weights in the upsampling layers of the V-Net architecture. These distributions are then optimized to their final values through iterative training with a process known as Bayes by Backprop[59]. Bayes by Backprop introduces a physics-inspired cost function known as variational free energy, which consists of two terms. The first term is the Kullback–Leibler divergence, which measures the complexity of the learned distribution against the Gaussian prior distribution. The second term is the negative log-likelihood, which measures the error with respect to the training examples.

For training each model, we are faced with memory constraints imposed by GPUs, and we randomly sample uniformly sized subvolumes from the large CT scans to fit the model on the GPUs. For inference, we deduce on CT scan subvolumes (with some overlap) and stitch the results together to generate a complete segmentation prediction, as in LaBonte et al.[46]. We generate 48 such predictions for each CT scan (Fig. 2f). Nominal predictions are calculated by turning off dropout in the MCDN and using the mean value of all weights in the BCNN.

Neural network training and inference were performed on two NVIDIA V100 GPUs with 32GB memory each.

**Probability maps and segmentation uncertainty**. Consider a collection of $N$ image segmentation samples of a 3D image in the set of classes $C = \{1, 2, ..., n_c\}$. The probability map, $\epsilon_{v,i}$ for voxel $v$ and class $i$ is calculated from these $N$ image segmentation samples using

$$\epsilon_{v,i} = \frac{1}{N}\sum_k^N p_{v,i}^k. \tag{1}$$

Here, $p_{v,i}^k$ is the binarized value of voxel $v$ in class $i$ in the $k$th image segmentation sample, where $p_{v,i}^k = 1$ if $v$ is in the segmented class and $p_{v,i}^k = 0$ if not. For a binary image ($n_c = 2$) we only consider a single probability map for class $i = 1$, $\epsilon_v$.

Segmentation uncertainty is quantitatively defined from the probability maps using the Shannon entropy measure:

$$H(\epsilon_v) = -\sum_{i \in C} \epsilon_{v,i} \log_2(\epsilon_{v,i})/\log_2(n_c), \tag{2}$$

where $\log_2(n_c)$ is utilized to normalize the uncertainty between 0 and 1. After normalization, a Shannon entropy equal to 1 indicates that the voxel is significantly uncertain, whereas values close to 0 are highly certain. Moreover, we can also quantify the segmentation uncertainty on a per-class basis using

$$H(\epsilon_{v,i}) = -\epsilon_{v,i}\log_2(\epsilon_{v,i})/\log_2(n_c). \tag{3}$$

Additional in-depth descriptions and demonstrations of the EQUIPS workflow for multi-class images ($n_c > 2$) can be found in the Supplementary Information.

**Discretization.** For the woven composite and medical exemplars, our physics simulation code requires a surface-conformal 3D volumetric mesh (discretization) of the simulation domain. First, a surface mesh is created from the images using the Lewiner marching cubes algorithm implemented in Python's scikit-image. This algorithm is applied directly to the probability map using a contour level set equal to the percentile threshold value. By applying the marching cubes algorithm to the real-valued probability map, the generated surface mesh is smooth rather than the stair-stepped mesh that would result from applying to a segmented data set. The resulting surface mesh is exported to the Standard Tessellation Library (STL) format.

We create volumetric meshes using the conformal decomposition finite-element method (CDFEM)[60], implemented in Sandia's Sierra/Krino code. First, a background tetrahedral mesh of the entire simulation domain is created at the desired mesh resolution using Cubit 15.5. Next, the STL file surface mesh is overlaid on the background mesh and decomposed using CDFEM.

**Woven composite.** A compression-molded silica phenolic composite was used for imaging. The woven composite material is composed of multiple layers of an 8-harness silica fiber cloth (Refrasil) impregnated with a phenolic resin (SC-1008, Durite) and pressed/cured according to Durite manufacturing specifications. It was imaged via X-ray computed tomography at a 7.3 micron resolution with a domain size of $616 \times 616 \times 979$ voxels.

For segmentation, the BCNN was trained using one full CT scan example with a manually generated label. We divided the scan into 64 subvolumes. For training, we used 54 subvolume examples and held 10 for model validation. The model was trained for five epochs with a learning rate of 0.001, and used $64 \times 128 \times 128$ voxel subvolumes of inputs, with a batch size of 8. The BCNN was given a zero mean, unit variance Gaussian prior and a Kullback–Leibler loss coefficient introduced at epoch 2, increasing by 0.25 per epoch thereafter.

Effective thermal conductivity is calculated by solving the steady-state Fourier's Law in three dimensions. The matrix is isotropic with thermal conductivity of $0.278 \, \text{W/(m} \cdot \text{K)}$, while the fabric material is transversely isotropic with thermal conductivity of $4.0 \, \text{W/(m} \cdot \text{K)}$ in the fabric in-plane direction with half that in the fabric-normal direction. A temperature gradient is applied in the out-of-plane direction using Dirichlet boundary conditions, and there is zero flux through all other boundaries. The effective thermal conductivity is calculated from the mean heat flux through one of the boundaries divided by the imposed temperature gradient.

Permeability of the fabric is calculated by solving for Stokes flow around the fabric phase (assuming the matrix is unfilled). A pressure gradient is imposed in the out-of-plane direction with no-slip boundary conditions imposed on the fabric surfaces and no-flux boundary conditions on the other external boundaries. Permeability is calculated from the resulting fluid flux on an external boundary, the porosity, and the imposed pressure gradient.

All physics simulations are performed using Sandia's Sierra/Aria Galerkin finite element code.

**Graphite electrodes in lithium-ion batteries.** E1 and E2 are commercially available Lithium-ion battery electrodes imaged using X-ray computed tomography. In particular, E1 and E2 represent Electrode IV[13] and Litarion[26] graphite datasets, respectively. E1 has a voxel size of 0.325 microns and an image size of $1100 \times 1100 \times 194$, while E2 has a voxel size of 0.1625 microns and an image size of $1100 \times 1100 \times 405$.

We trained two BCNN models, one with each of the E1 and E2 battery examples and validated each model using the same examples, but flipped along the $x$-axis with manually generated labels. Each model was trained for three epochs with a learning rate of 0.001. The inputs in one batch size consisted of $88 \times 176 \times 176$ voxel subvolumes. The BCNN used a zero-mean-unit-variance Gaussian prior and a Kullback–Leibler loss coefficient that started at 0.33. After each epoch, the loss coefficient increased by 0.33.

An in-house finite-volume method code developed by Mistry et al.[61] solves Laplace's equation for tortuosity and electrical conductivity separately, using a structured voxellated grid (implying that the discretization step described above is not necessary). Simulations involving E1 and E2 are performed on a subdomain size of $84.5 \times 84.5 \times 68.575$ and $84.5 \times 84.5 \times 65.8125 \, \mu m$, respectively. Laplace's equation is solved for both tortuosity and electrical conductivity, with a nondimensional conductivity value of unity for the transporting phase (pore space for tortuosity, particle phase for conductivity) and a value of $10^{-6}$ for the non-transporting phase. A potential gradient is applied using Dirichlet boundary conditions on opposing boundaries with no flux on the remaining boundaries. The effective transport property is calculated by dividing the resulting flux by the imposed potential gradient. Similarly, tortuosity is calculated as the porosity over the effective transport property.

**Human torso.** Our medical data is taken from an open-source database of anonymous patient medical images consisting of 3D CT scans and manual segmentations of the chest/abdominal region[62]. Scan 2.2 is used for this exemplar, which has a dimension of $512 \times 512 \times 219$ voxels and per-voxel resolution $0.961 \times 0.961 \times 2.4$ mm.

The MCDN is used for this exemplar because of its ability to perform multi-class (i.e. multi-organ) segmentation, whereas the BCNN currently supports one-class binary segmentation (suitable only for a single organ/material). The model is trained using two labeled examples (scans 2.1 and 2.2) that were normalized and transformed into logarithmic space. Six subvolumes of $64 \times 192 \times 192$ voxels were used for each image, four down- and up-sampling blocks in the V-Net architecture, and a dropout rate of 0.1.

Effective Young's modulus in the axial (vertical) direction is calculated by solving for the quasi-static conservation of linear momentum using Sandia's Sierra/Aria Galerkin finite-element code. The bone is considered to be a linear elastic material with bulk modulus of 4.762 GPa and Poisson's ratio of 0.22. The aorta and other surrounding tissues are omitted from the simulation. The top of the spine is held fixed in space, while a prescribed vertical displacement of $10^{-4}$ m is applied to the bottom spine surface. Young's modulus is calculated using the normal force on the bottom surface, the applied displacement, and the domain length.

Aortic blood flow simulations were performed using Sierra/Fuego. The incompressible Navier–Stokes equations were solved using a Newtonian constitutive model with dynamic viscosity $\mu = 0.003 \, \text{kg} \cdot \text{m}^{-1} \cdot \text{s}^{-1}$ and blood density $\rho = 1060 \, \text{kg} \cdot \text{m}^{-3}$[63]. No-slip boundary conditions are enforced on the aorta walls and zero-pressure boundary conditions at the outlet surfaces. The inflow pressure is transient to mimic physiological circulation patterns and is estimated from Benim et al.[63] using the Hagen–Poisuille equation to achieve a time-averaged volumetric flow rate of 8 L/min. Simulations are initialized with a stationary fluid and are performed for 10 complete pulses to achieve a pseudo-steady-state flow profile.

We calculate two physics quantities from these aorta simulations. First is the flow-rate ratio, which is the ratio of the flow rate through the smaller side vessels branching off the main abdominal aorta to the flow rate through the main aorta branch vessels. The second is the wall-shear-stress threshold ratio $\delta_\tau$. We compute the wall shear stress on the aorta walls and identify any area with a stress measurement above 0.2 Pa, which indicates a significant risk for aneurysm[50]. This value is then normalized by the total aorta surface area for the physics quantity that we investigate. The aortic wall shear stress is relevant as an indicator for aneurysms[50]. Thus, this wall-shear-stress threshold ratio indicates the percentage of the aorta wall vulnerable to aneurysms.

## Data availability

The probability maps and simulation results for each of the exemplar problems has been deposited in the Mendeley Data database at https://doi.org/10.17632/g3hr4rkb48[64]. Probability maps are available as Numpy arrays and each physics quantity uncertainty distribution is available as a CSV file.

## Code availability

The Bayesian Convolutional Neural Network (BCNN) source code is available on GitHub: https://github.com/sandialabs/bcnn[65]. The Monte Carlo Dropout Network (MCDN) source code is available on GitHub: https://github.com/sandialabs/mcdn-3d-seg[66]. A python Jupyter notebook demonstrating the entire EQUIPS workflow on a simple manufactured image is available[64].

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

## Acknowledgements

The authors gratefully acknowledge contributions and discussions from the Credible, Automated Meshing of Images project team. Specifically we are appreciative to Kevin Potter and Matthew Smith for fruitful discussions and input on the use of machine learning for image segmentation and uncertainty quantification, Benjamin Schroeder for uncertainty quantification insights, and David Noble for assistance with the mesh generation and discretization algorithms. We acknowledge Francesco Panerai for the CT images of the woven composite material. Jacquilyn Weeks provided instructive feedback about the manuscript clarity, organization, and story. We finally acknowledge Dan Bolintineanu, Kyle Neal, and Benjamin Schroeder for pre-publication peer review. This work was supported by the Laboratory Directed Research and Development program at Sandia National Laboratories, a multimission laboratory managed and operated by National Technology and Engineering Solutions of Sandia, LLC, a wholly owned subsidiary of Honeywell International, Inc., for the U.S. Department of Energy's National Nuclear Security Administration under contract DE-NA-0003525. This paper describes objective technical results and analysis. Any subjective views or opinions that might be expressed in the paper do not necessarily represent the views of the U.S. Department of Energy or the United States Government.

## Author contributions

M.C.K. acquired, analyzed, and interpreted data, designed the workflow, and substantially wrote the manuscript. T.L., C.M., C.N., and K.S. acquired data, created software used in the work, and contributed to the manuscript. L.N.C. designed the workflow, created software used in the work, and contributed to the manuscript. P.P.M. supervised the work and contributed to the manuscript. S.A.R. conceived and designed the workflow, analyzed and interpreted data, and contributed to the manuscript.

## Competing interests

The authors declare no competing interests.
