## [Peer Review File · Nature Communications]

Quantifying the unknown impact of segmentation uncertainty on image-based simulationsReviewers' Comments:

Reviewer #1:

Remarks to the Author:

This manuscript develops and demonstrates a workflow for the propagation of uncertainty embedded in imagery into the uncertainty in outcomes of physics simulations on geometries based on those images. In particular, a workflow is demonstrated making use of convolutional neural networks to create binary segmentations of X-ray CT imagery of different materials; the uncertainty in the identification of parts of the image into one phase or the other are generated from Monte Carlo or Bayesian Convolutional Neural Networks resulting in a probability map for the image. The probability map is used to create realisations of the image at select probability thresholds on which simulations are run. Variation in the outcome of the simulations shows the propagation of the image uncertainty into the "physics uncertainty".

This type of framework is much needed in the ever increasing body of research making use of simulations on imagery acquired from X-ray imaging. The application to three distinct material types and distinct physical processes shows it is versatile and general. I am sure the paper would have widespread impact; it is a simple enough of an approach, and the clear next step in developing rigour into this type of analysis. The paper is very well written.

Below are three critical comments which I think can be addressed without too much effort:

1. The key to the approach is the creation of probability maps. The novelty in this paper, however, is not in the creation of the probability map but rather the next step of using that map to generate a distribution of simulation outcomes. To that end, I would expect to see far more material in the generation of the distribution of simulation outcomes. In practice, it may be useful at times to use the fewest number of simulations in combination with an assumed curve. But, in demonstrating the approach, and its use for non-trivial distributions, it is important to generate an accurate outcome distribution model. Why are so few simulations run? It would be far more insightful, particularly for, e.g., the permeability calculation and the Young's modulus of the torso to illustrate the full distribution in detail; also not as a CDF.

2. Related to this, it is stated that the approach is general to any approach to segmentation so long as some method for generating a probability map is created. Yet the authors already point out the problems with creating probability maps from Monte Carlo Convolutional Networks. Surely there will be significant problems with other approaches, particularly with manual segmentation which is still in widespread use. It would strengthen the work to identify whether there exist accurate approaches to generating probability maps from these approaches. Their absence seems to preclude the broad use of this technique as suggested. Perhaps we are all headed towards the use of CNN or Weka anyway, but if so, this should be argued.

3. The approach is only demonstrated for binary segmentation. The authors should comment on how this might be developed for systems with greater than 2 phases of interest. Multiphase fluid flow through rocks comprises at least 3 phases (2 fluid and 1 rock); if multiple minerals are of interest there can be many more.

Reviewer #2:

Remarks to the Author:

Summary

The work concerns uncertainty estimation in the task of image-based simulation. More specifically, the Authors focus on segmentation using CNNs, which can be sampled either using dropout (referred to as MCDNs) or from estimated network weight distributions (BCNNs). In a proposed procedure called EQUIPS, from these sampled segmentations, a distribution of physical quantities of interest can be estimated. Due to the uncertainty of the segmentations, the physical quantity estimates are also uncertain. The EQUIPS procedure provides CDFs of the physical quantity estimates at output.

The paper is well written and self – contained. I highly appreciate three versatile exemplar applications. Some crucial notions, however, are not defined. The procedure as described is inefficient. It is also not clear how well it generalizes to problems with many classes (segmentation which is not a binary classification problem).

Major comments

1. The description lacks a definition of two central notions: the “uncertainty” and “probability map”. These two notions are crucial for the EQUIPS procedure. For example: in the Figure 2, providing the overview of the approach it writes: “In EQUIPS (bottom), segmentation uncertainty is calculated by creating many Monte Carlo image segmentations (f) and combining them into a probability map (g).” It is confusing for the reader that the terms quantification of uncertainty and probability maps occur multiple times in the text, but remain unclear.

In fact, it seems that the actual uncertainty of image segmentation is not quantified at all. Indeed, multiple samples of segmentation are obtained, but there isn’t any quantity computed which corresponds to image segmentation uncertainty. In contrast, uncertainty of the output physics quantities is obtained as an estimate of the estimator distribution.

2. Most importantly, probability maps are said to be computed but it is not stated how. This is specifically problematic to readers who are familiar with the literature in uncertainty estimation using dropout.

Specifically, the work by Gal (ref.

Smith, L. & Gal, Y. Understanding measures of uncertainty for adversarial example detection.

CoRRabs/1803.08533 (2018) and Gal, Y. Dropout as a bayesian approximation: Representing model uncertainty in deep learning. Proceedings of the 33rd International Conference on Machine Learning (2016).), defined the measures of uncertainty denoted entropy and BALD.

When the probability maps are computed in the submitted work based on the monte-carlo (dropout) approach, are they computed based on these measures?

An alternative way would be simply to for each voxel take directly the probability of the class in each sample segmentation (Fig 2f) and then assign the voxel to the most probable class, and then compute the probability map using voting (as fraction of the samples that assigned to that class). There could be also a number of other ways. So it would be great if the Authors gave a formula for the probability maps.

3. On top of that Authors also define uncertainty maps, which are based on probability maps (line 157). These maps give nice visualizations but again since their definition depends in turn on probability maps, it would be fantastic to introduce them formally and as part of the EQUIPS, and put in Figure 2.

4. The described EQUIPS procedure seems inefficient. Assuming that the final target is the estimation of the CDF for the physical quantities, it is unclear why the the probability maps need to be computed for many percentiles up-front. If the output distribution does fit the Gaussian, only 3 percentiles are needed to model it. So more efficient would be to have some loop which takes only a few percentiles, gives the first estimate based on the Gaussian, and only after it turns out that the distribution does not fit to the Gaussian other percentiles need to be computed. Finally, there is no guidance given what to exactly do if the distribution does not look like Gaussian. Ideally, the computation of the other percentiles and the search for the family would be somehow automatized. This would make the approach both more efficient and more realistic, as the Gaussian assumption is not satisfied for many cases in the exemplars given by the Authors.

5. Finally, it is not clear how this procedure generalizes to multiple classification problems, i.e., segmentation where the regions could be of many types. First, such boundary regions as visualized in Fig 1 would not be enough, as in such cases it is quite likely the entire region is differently classified by the different sampled networks. Also, how exactly are the probability maps computed and uncertainty estimated? Again, entropy and BALD work naturally for multiple classes. Here it is unclear. Also, the authors mention they solve it by solving multiple binary problems instead. It is not sure whether this approach does not underestimate the uncertainty.

Minor comments

1. In the Introduction, the Authors claim that

„uncertainty through stochastic sampling of dropout layers is an approach with questionable statistical validity“, and that

“LaBonte et al. 46 has developed a Bayesian CNN (BCNN) that measures uncertainty in the weight space, resulting in statistically-justified segmentation uncertainty quantification.”

First of all, the reference 46 does not point to a published paper. Instead, it lists the co-authors of the submitted work, a title and a year. It does not suffice to back-up the claims. Without justifying these claims, they should be dropped from the paper altogether. If the Authors choose to keep the claims, for the claims to be justified, concrete arguments need to be provided. Specifically, why is the dropout approach questionable, and why is the bcnn justified? Finally, if the authors find the dropout approach questionable, why do they use it in EQUIPS in the end?

Response to Reviewer 1

We would like to thank the referee for the careful review of our manuscript and for the insightful comments provided. We have carefully considered your comments and modified the manuscript in response to each comment. Below we exactly replicate and respond to each comment in blue. A separate version of the manuscript with “track changes” is available to highlight the improvements to the manuscript. We hope that the referee is satisfied with our responses to their comments and the corresponding improvements to the manuscript.

1. **Comment:** The key to the approach is the creation of probability maps. The novelty in this paper, however, is not in the creation of the probability map but rather the next step of using that map to generate a distribution of simulation outcomes. To that end, I would expect to see far more material in the generation of the distribution of simulation outcomes. In practice, it may be useful at times to use the fewest number of simulations in combination with an assumed curve. But, in demonstrating the approach, and its use for non-trivial distributions, it is important to generate an accurate outcome distribution model. Why are so few simulations run? It would be far more insightful, particularly for, e.g., the permeability calculation and the Young’s modulus of the torso to illustrate the full distribution in detail; also not as a CDF.

Response: Thank you for your comment. It is worth noting that once we have a probability map, we do not *need* a large number of samples, because we do not have to *build* a distribution. Traditionally, a probability distribution function (PDF, or its integrated cousin the CDF) is generated by taking a large number of independent samples of the uncertain space and constructing a histogram or a kernel density estimate (KDE). Getting an accurate representation of this distribution requires a large number of samples. However, this is the process used to generate the probability map, where in this paper, we took 48 Monte Carlo samples through the CNN to generate the probability map.

Once you have your representation of the uncertainty distribution (traditionally the KDE, PDF, or CDF – in our case the probability map), one does not necessarily need a large number of points from that distribution to reconstruct the distribution because you are not sampling it randomly, but instead systematically at certain percentiles. This is why, in most cases, we can get by with using as few as three points from the probability map to construct the uncertainty distribution in the physics quantity. This works when the eventual distribution is Normal. However, one does not know that the distribution will necessarily be Normal until more points are calculated. This is why, for most of the physics simulations, we performed 10 simulations to fully show the shape of the distribution. We believe that 10 points are sufficient to show whether a distribution is likely Normal or not. Once the 10 points show a Normal distribution, 3 points are sufficient for replicating that on another image of the same type and the same physics simulation. This is one of the novel points of this paper, that, once established for a particular exemplar, the EQUIPS approach does not require large numbers of expensive physics simulations to estimate physics quantity uncertainty.

That said, the reviewer makes a very good point about the number of samples in the non-trivial distributions towards the end of the paper, particularly Young’s modulus of the spine. We agree that there were not sufficient points in that exemplar to fully resolve the distribution. We have updated Figure 5(e) to include 25 samples, with more points concentrated near the inflection point of the distribution.

We would like to emphasize that the purpose of the manuscript is to shed light on segmentation

uncertainty and its effect on image-based physics simulations; and not on high resolution uncertainty distributions for any particular exemplar. Instead, we focus the intention of the paper on the EQUIPS workflow and demonstrating its ability to systematically propagate segmentation uncertainty to physics simulations using a diverse set of examples. A researcher adopting the EQUIPS workflow could feel free to use as many samples to resolve the distribution as they desire.

2. **Comment:** Related to this, it is stated that the approach is general to any approach to segmentation so long as some method for generating a probability map is created. Yet the authors already point out the problems with creating probability maps from Monte Carlo Convolutional Networks. Surely there will be significant problems with other approaches, particularly with manual segmentation which is still in widespread use. It would strengthen the work to identify whether there exist accurate approaches to generating probability maps from these approaches. Their absence seems to preclude the broad use of this technique as suggested. Perhaps we are all headed towards the use of CNN or Weka anyway, but if so, this should be argued.

Response: The most accurate approach to generating a probability map is one where the method that produces the image segmentation samples is unbiased by human decisions. For this reason, we primarily use the BCNN approach that we have developed. While it is far from perfect, it does provide a somewhat objective calibration of model and image uncertainty through the Bayesian method of calibrating both weights and variances in the neural network. Similarly, algorithms like Weka objectively produce probability maps that are unbiased by human interaction, other than through the manual segmentation image data on which they were trained. At this point, the authors know of no better way to generate less biased probability maps, and we certainly do believe that tools like CNNs and Weka are the way of the future for image segmentation.

That said, it is clear that manual segmentation approaches are extremely widespread in practice and are unlikely to disappear overnight. While it is certainly not ideal, it is possible to generate probability maps by having many subject matter experts manually segment an image, then combining them in the same way that we do with the CNN-based image segmentation samples (see the new mathematical description of this approach beginning on line 432 of the revised manuscript). However, to get a good statistical representation of the probability map (yet still biased by human interaction) with manual segmentation would require tens of expert segmentations. As requested, we have added to our discussion on this approach in the Discussion section, beginning around line 356. Note also that in response to this comment, we have renamed “Monte Carlo image segmentations” to “image segmentation samples” in recognition that there may be many segmentation approaches used to generate the probability map, and not all of them involve Monte Carlo sampling.

3. **Comment:** The approach is only demonstrated for binary segmentation. The authors should comment on how this might be developed for systems with greater than 2 phases of interest. Multiphase fluid flow through rocks comprises at least 3 phases (2 fluid and 1 rock); if multiple minerals are of interest there can be many more.

Response: This is a very insightful comment by the reviewer, and we also received a similar request from the other reviewer. While we did demonstrate a multi-class segmentation in the section on the human torso, the resulting physics simulations were entirely binary, relying only on the geometry from a single class. To further generalize the EQUIPS approach to true multi-class segmentations and image-based simulations, we have added an entirely new

supplemental information section that explains how EQUIPS could be applied to a multi-class segmentation in detail. We have also briefly discussed this in the Discussion section with a new paragraph beginning on line 374.

Response to Reviewer 2

We would like to thank the referee for the careful review of our manuscript and for the insightful comments provided. We have carefully considered your comments and modified the manuscript in response to each comment. Below we exactly replicate and respond to each comment in blue. A separate version of the manuscript with “track changes” is available to highlight the improvements to the manuscript. We hope that the referee is satisfied with our responses to their comments and the corresponding improvements to the manuscript.

1. **Comment:** The description lacks a definition of two central notions: the “uncertainty” and “probability map”. These two notions are crucial for the EQUIPS procedure. For example: in the Figure 2, providing the overview of the approach it writes: “In EQUIPS (bottom), segmentation uncertainty is calculated by creating many Monte Carlo image segmentations (f) and combining them into a probability map (g).” It is confusing for the reader that the terms quantification of uncertainty and probability maps occur multiple times in the text, but remain unclear.

In fact, it seems that the actual uncertainty of image segmentation is not quantified at all. Indeed, multiple samples of segmentation are obtained, but there isn’t any quantity computed which corresponds to image segmentation uncertainty. In contrast, uncertainty of the output physics quantities is obtained as an estimate of the estimator distribution.

Response: We appreciate this insightful comment from the reviewer on the clarity and definition of some of our nomenclature, and agree that there was some ambiguity and interchanging of the phrases “segmentation uncertainty” and “probability map” in the manuscript, leading to confusion.

As for the *probability map*, what the reviewer quoted was from the caption of Figure 2. A more precise definition of the probability map is found in the second paragraph of the results section, beginning at line 112. However, to improve the clarity of these terms, we have added a new Methods subsection (beginning on line 432) that includes a more precise mathematical definition of the probability map.

We introduce the concept of *segmentation uncertainty* at line 43 of the manuscript, highlighting it visually in Figure 1(b). In this qualitative example, the yellow shaded region is the “uncertain” portion of the segmentation. However, we agree with the reviewer that its precise definition was unclear. In response, we have adopted a more common definition of uncertainty, the *Shannon entropy* and updated the uncertainty map in Figure 3(a) to use this quantitative value. Additionally, we have included its precise definition in the new Methods subsection.

It is worth noting that the segmentation uncertainty, as quantified by the Shannon entropy, is a useful quantity for visualizing and understanding uncertainty, but it is not a required part of the EQUIPS workflow. Instead, all of physics calculations rely solely on probing the probability map, which is directly used to calculate the segmentation uncertainty. It is for this reason that probing the probability map is the same as probing segmentation uncertainty.

2. **Comment:** Most importantly, probability maps are said to be computed but it is not stated how. This is specifically problematic to readers who are familiar with the literature in uncertainty estimation using dropout. Specifically, the work by Gal (ref. Smith, L. & Gal, Y. Understanding measures of uncertainty for adversarial example detection. CoRRabs/1803.08533 (2018) and Gal, Y. Dropout as a bayesian approximation: Representing model uncertainty

in deep learning. Proceedings of the 33rd International Conference on Machine Learning (2016).), defined the measures of uncertainty denoted entropy and BALD. When the probability maps are computed in the submitted work based on the monte-carlo (dropout) approach, are they computed based on these measures? An alternative way would be simply to for each voxel take directly the probability of the class in each sample segmentation (Fig 2f) and then assign the voxel to the most probable class, and then compute the probability map using voting (as fraction of the samples that assigned to that class). There could be also a number of other ways. So it would be great if the Authors gave a formula for the probability maps.

Response: The reviewer’s suggested calculation for the probability map is precisely the approach that we take in the manuscript. As discussed in our response to comment 1, we have provided a more precise definition for the probability map in the new Methods subsection.

As for the measures of uncertainty using entropy (predictive entropy) and Bayesian Active Learning by Disagreement (BALD), neither are directly applied here. With EQUIPS, the creation of the probability map is independent of the CNN and, as a result, these uncertainty measures are not directly applicable to image segmentation uncertainty quantification. Thus, we do not use the entropy or BALD to *create* the probability map. Yet, the reviewer has a point that entropy is a more fundamental and physical measurement of uncertainty. Thus, at the suggestion of the reviewer, we’ve shifted to describing the uncertainty map using the Shannon entropy. The original definition of the uncertainty map was arguably somewhat arbitrary while the Shannon entropy is not. We thank the reviewer for this suggestion that led to an improvement in the manuscript.

3. **Comment:** On top of that Authors also define uncertainty maps, which are based on probability maps (line 157). These maps give nice visualizations but again since their definition depends in turn on probability maps, it would be fantastic to introduce them formally and as part of the EQUIPS, and put in Figure 2.

Response: Thank you for this suggestion. As noted in more detail in our response to comment 1, we have added improved and precise definitions of the segmentation uncertainty to the new Methods subsection. Because the uncertainty maps are not directly used in any calculation and are instead for visualization, we have omitted it from the EQUIPS workflow of Figure 2.

4. **Comment:** The described EQUIPS procedure seems inefficient. Assuming that the final target is the estimation of the CDF for the physical quantities, it is unclear why the the probability maps need to be computed for many percentiles up-front. If the output distribution does fit the Gaussian, only 3 percentiles are needed to model it. So more efficient would be to have some loop which takes only a few percentiles, gives the first estimate based on the Gaussian, and only after it turns out that the distribution does not fit to the Gaussian other percentiles need to be computed. Finally, there is no guidance given what to exactly do if the distribution does not look like Gaussian. Ideally, the computation of the other percentiles and the search for the family would be somehow automatized. This would make the approach both more efficient and more realistic, as the Gaussian assumption is not satisfied for many cases in the exemplars given by the Authors.

Response: Thank you for pointing out this unclear portion of our manuscript. There are two aspects to the EQUIPS workflow that must be separately addressed. The first is the generation of the probability map, and the second is the interrogation of that probability

map to perform physics simulations.

First, the generation of the probability map involves taking *independent* and essentially *random* samples of the segmentation approach. Generating the probability map requires *building* a distribution stochastically, essentially like creating a histogram. This necessarily requires a large number of samples, otherwise the estimate of the standard deviation of that distribution would be high and the confidence interval of the mean value would be large. We see no way to generate statistically significant probability maps with very few random image segmentation samples. We have added precise definitions of the probability map to the new Methods subsection to make this more clear. Fortunately, performing many inferences with our CNN approach is relatively efficient, so the requirement to have many samples to create a probability map is not prohibitive. Of course, doing so with manual segmentations would be very time consuming, as we newly discuss in the manuscript.

Secondly, in calculating the uncertainty of physics quantities, where the physics simulations are possibly more computationally expensive, the reviewer’s suggestion on efficient use of the EQUIPS procedure is *exactly* what we propose in the manuscript. In particular, the first portion of the results section, beginning in the third paragraph, line 123, describes our recommended approach. Briefly, we recommend beginning with the three “standard segmentations”, which would fully describe the results if they followed a Normal distribution. We then recommend adding a few points at a time to then test the hypothesis that the results follow a Normal distribution. For the majority of the exemplars, we study 10 additional percentile segmentations to confirm the form of the distribution. However, once the Normal form is confirmed, only the three standard segmentations would be required for future applications of EQUIPS to the same type of image and the same physics simulation. For more non-trivial physics quantity uncertainty distributions, it is highly likely that more than three percentile segmentation simulations will be required. This is all discussed in detail beginning on line 325 of the Discussion section.

One could imagine an automated algorithm to iteratively probe the probability map to generate the output physics quantity uncertainty distributions, iteratively fitting that data to a possible family of distributions. While this would certainly be a worthwhile endeavor for a specific exemplar, developing such an automated, generalized, iterative algorithm is beyond the scope of this work.

5. **Comment:** Finally, it is not clear how this procedure generalizes to multiple classification problems, i.e., segmentation where the regions could be of many types. First, such boundary regions as visualized in Fig 1 would not be enough, as in such cases it is quite likely the entire region is differently classified by the different sampled networks. Also, how exactly are the probability maps computed and uncertainty estimated? Again, entropy and BALD work naturally for multiple classes. Here it is unclear. Also, the authors mention they solve it by solving multiple binary problems instead. It is not sure whether this approach does not underestimate the uncertainty.

Response: This is a very insightful comment by the reviewer, and we also received a similar request from the other reviewer. While we did demonstrate a multi-class segmentation in the section on the human torso, the resulting physics simulations were entirely binary, relying only on the geometry from a single class. To further generalize the EQUIPS approach to true multi-class segmentations and image-based simulations, we have added an entirely new supplemental information section that explains how EQUIPS could be applied to a multi-class segmentation in detail. We have also briefly discussed this in the Discussion section with a

new paragraph beginning on line 374.

6. **Comment:** In the Introduction, the Authors claim that “uncertainty through stochastic sampling of dropout layers is an approach with questionable statistical validity”, and that “LaBonte et al. 46 has developed a Bayesian CNN (BCNN) that measures uncertainty in the weight space, resulting in statistically-justified segmentation uncertainty quantification.” First of all, the reference 46 does not point to a published paper. Instead, it lists the co-authors of the submitted work, a title and a year. It does not suffice to back-up the claims. Without justifying these claims, they should be dropped from the paper altogether. If the Authors choose to keep the claims, for the claims to be justified, concrete arguments need to be provided. Specifically, why is the dropout approach questionable, and why is the bcnn justified? Finally, if the authors find the dropout approach questionable, why do they use it in EQUIPS in the end?

Response: We apologize for the incomplete reference. The cited paper is a preprint published openly on arXiv. The reference has been updated with the full citation information.

Regarding the comment about the credibility of the dropout approach, we have updated our discussion around line 68 to more adequately reflect our intent. Reference 45 does include discussion about the credibility of Monte Carlo dropout. We believe that the BCNN approach to generating probability maps is superior to Monte Carlo dropout, and use it primarily throughout the manuscript. However, this is a relatively new technique, with limited open source code available when compared to other segmentation techniques. It is our intent to show the generality of the EQUIPS workflow in this manuscript. For this reason, we strived to illustrate the workflow without limiting it to the BCNN, but also with more common Monte Carlo dropout codes, and in addition, other tools such as Weka and manual segmentation.

Reviewers' Comments:

Reviewer #1:

Remarks to the Author:

The authors have addressed my concerns

Reviewer #2:

Remarks to the Author:

In principle, I find the rebuttal letter as addressing my concerns in a satisfying manner. The main text, however, was not updated nor improved enough.

In more detail, although the Methods now include some useful definitions, I still find the main text of the paper written in a very ambiguous way, leaving a large room for interpretation and doubts about what is actually being computed and how. The paper would be much easier to read if precise nomenclature was kept and if quantitative interpretation wasn't overflowing the facts.

For example, the word "uncertainty" appears in the text 149 times, "uncertainties" 3 times, and most of the time it is not really clear what exactly the Authors mean by it.

Another example is the notion of "uncertainty distribution". If uncertainty is defined as the Shannon entropy, it is hard to see why there are plots of CDF of various quantities such as Relative Tortuosity, and they are called "uncertainty distribution". In contrast, the actual entropy is never visualized. Also, if uncertainty is formally quantified by the entropy, what would actually correspond to the uncertainty distribution?

Response to Reviewer 2

We are happy that our original response to your comments satisfied most of your concerns. However, we do see that you still have one concern remaining. Below we exactly replicate and respond to this comment in blue. We hope that the referee is satisfied with our response to your comments.

Comment: In principle, I find the rebuttal letter as addressing my concerns in a satisfying manner. The main text, however, was not updated nor improved enough.

In more detail, although the Methods now include some useful definitions, I still find the main text of the paper written in a very ambiguous way, leaving a large room for interpretation and doubts about what is actually being computed and how. The paper would be much easier to read if precise nomenclature was kept and if quantitative interpretation wasn't overflowing the facts.

For example, the word "uncertainty" appears in the text 149 times, "uncertainties" 3 times, and most of the time it is not really clear what exactly the Authors mean by it.

Another example is the notion of "uncertainty distribution". If uncertainty is defined as the Shannon entropy, it is hard to see why there are plots of CDF of various quantities such as Relative Tortuosity, and they are called "uncertainty distribution". In contrast, the actual entropy is never visualized. Also, if uncertainty is formally quantified by the entropy, what would actually correspond to the uncertainty distribution?

Response: We acknowledge your confusion with the numerous uses of the word "uncertainty" in our manuscript. It is, unfortunately, unavoidable given that the entire manuscript pertains to the quantification and propagation of uncertainty. We would like to emphasize that we very rarely use "uncertainty" by itself. Instead, it occurs as part of multi-word phrases. Each phrase has a specific meaning that is well defined in the manuscript, and we have tried our hardest to use all of them consistently.

"Segmentation uncertainty" is the concept that one can segment an image differently, resulting in different segmentations. This is precisely defined on line 43 of the manuscript, including the entire surrounding paragraph, and is qualitatively illustrated in Figure 1.

"The range of possible segmentations, accounting for all tools and variables, represents the range of image *segmentation uncertainty* (the yellow regions in Figure 1(b)), within which the correct answer will be found."

While this phrase is really a big-picture concept, it can also be quantified using the Shannon entropy (precisely defined on line 434 of the methods section). Despite the reviewer's comment that it is never visualized, it is shown in Figure 3(a). However we should note that this quantification of segmentation uncertainty using the Shannon entropy is not used in the EQUIPS workflow itself; it is only a handy tool for visualizing regions of the image where the segmentation uncertainty is high.

On the other hand, a physics quantity can have an "uncertainty distribution" that is the result of the propagation of segmentation uncertainty through physics simulations. This uncertainty distribution, which is precisely defined on line 74 of the manuscript, quantifies the impact of segmentation uncertainty on the physics quantities of interest (i.e., tortuosity, permeability, thermal conductivity, and etc.).

"In this paper, we demonstrate that segmentation uncertainty leads to an *uncertainty distribution* in physics quantities predicted by image-based simulations."

and

“Herein, we address this challenge by presenting a systematic method of quantifying segmentation uncertainty and propagating that uncertainty through image-based simulations to create uncertainty distributions on predicted physics quantities ...”

We have consistently used the nomenclature “uncertainty distribution” with this meaning throughout the paper. Calculating the physics quantity uncertainty distribution is the primary goal of the manuscript and is the output of the EQUIPS workflow.

The uncertainty distribution is consistently shown throughout the manuscript as a cumulative distribution function (CDF). From the manuscript:

“This characteristic CDF estimates the uncertainty distribution of a physics quantity as a result of segmentation uncertainty.”

Each physics simulation is performed on an image segmentation generated by probing the probability map at a specific percentile value and results in a single value of a physics quantity. This calculated value represents that specific percentile value in a CDF. This process is repeated multiple times at different percentile values to fully populate the CDF. The resulting populated CDF is the uncertainty distribution for that physics quantity resulting from image segmentation uncertainty. A CDF (and its cousin the probability distribution function or PDF) is an incredibly common method of representing uncertainty in a physics quantity in the uncertainty quantification, verification, and validation community. We know of no better way of representing this data.

We have made a few minor changes to the manuscript to clarify phrasing. However, we have been unable to make significant modifications to the manuscript in response to your concerns, as it is not clear to us how to do so without addition clarity in review. We would be happy to improve the readability of the manuscript, but the reviewer has given no specific suggestions as to how it could be improved. Instead, we hope that this detailed response and the minor modifications to the manuscript clarifies your misunderstanding and satisfies your concerns.